# Prestige and homophily predict network structure for social learning of medicinal plant knowledge

Matthew O. Bond[1]*, Orou G. Gaoue[1,2,3,4]

1 Department of Botany, University of Hawai'i at Mānoa, Honolulu, Hawai'i, United States of America,
2 Department of Ecology and Evolutionary Biology, University of Tennessee at Knoxville, Knoxville, Tennessee, United States of America, 3 Faculty of Agronomy, University of Parakou, Parakou, Benin, 4 Department of Geography, Environmental Management and Energy Studies, University of Johannesburg, APK Campus, Johannesburg, South Africa

* mb2286@hawaii.edu

**Data Availability Statement:** The data sets supporting the results of this article are included within the article and its additional files or in the Dryad Digital Repository (doi:10.5061/dryad. cfxpnvx3q).

## Abstract

Human subsistence societies have thrived in environmental extremes while maintaining biodiversity through social learning of ecological knowledge, such as techniques to prepare food and medicine from local resources. However, there is limited understanding of which processes shape social learning patterns and configuration in ecological knowledge networks, or how these processes apply to resource management and biological conservation. In this study, we test the hypothesis that the prestige (rarity or exclusivity) of knowledge shapes social learning networks. In addition, we test whether people tend to select who to learn from based on prestige (knowledge or reputation), and homophily (e.g., people of the same age or gender). We used interviews to assess five types of medicinal plant knowledge and how 303 people share this knowledge across four villages in Solomon Islands. We developed exponential random graph models (ERGMs) to test whether hypothesized patterns of knowledge sharing based on prestige and homophily are more common in the observed network than in randomly simulated networks of the same size. We found that prestige predicts five hypothesized network configurations and all three hypothesized learning patterns, while homophily predicts one of three hypothesized network configurations and five of the seven hypothesized learning patterns. These results compare the strength of different prestige and homophily effects on social learning and show how cultural practices such as intermarriage can affect certain aspects of prestige and homophily. By advancing our understanding of how prestige and homophily affect ecological knowledge networks, we identify which social learning patterns have the largest effects on biocultural conservation of ecological knowledge.

## Introduction

Cultural evolution theory suggests that human adaptation depends on social learning, which is the ability to transmit information and ideas between individuals and accumulate knowledge

**Funding:** MOB was funded by a National Science Foundation Graduate Research Fellowship (https://www.nsfgrfp.org), the American Philosophical Society Lewis & Clark Fund for Exploration & Field Research (https://www.amphilsoc.org/grants/lewis-and-clark-fund-exploration-and-field-research), and the Garden Club of America Anne S. Chatham Fellowship in Medicinal Botany (https://www.gcamerica.org/scholarships/details/id/8). OGG was funded by a National Science Foundation Grant #1513354 (https://www.nsf.gov/index.jsp), and a start up grant from the University of Tennessee Knoxville (https://research.utk.edu/forms/start_up_funds/). The funders had no role in study design, data collection and analysis, decision to publish, or preparation of the manuscript.

**Competing interests:** The authors have declared that no competing interests exist.

over generations [1, 2]. People have adapted to diverse environments across the world since prehistoric times and now dominate the entire globe [3, 4]. The key to this global domination has been a network of biological and cultural adaptations [5, 6]. Humans have developed many cultural adaptations, such as intricate construction knowledge for technologies such as boats, complex practices to detoxify foods, and belief systems to encourage conservation [7–9]. To understand these and other cultural adaptations fundamental to nature conservation and the success of our species, we need to understand the evolution of social learning [10]. Here, we define social learning as knowledge acquired through instruction, observation, and other social interactions, which accumulates skills and information over time that no individual could develop in a single lifetime. In contrast, asocial learning is experiential and involves trial and error [10–12].

For social learning to produce an adaptive benefit, people must limit who they learn from; however, there is debate about how people choose who to learn from and how their choices affect social learning network structure [10–12]. Cultural evolution theory predicts that there is variation in knowledge, and knowledge that has a fitness benefit is more likely to be passed on to others [6]. Although social learning can increase fitness by saving time and energy, computer models show that fitness decreases as rates of social learning increase because the information learned becomes less accurate [10, 11]. Thus, to maximize their fitness humans must determine whether or not they should learn from another person [6]. The decision to use social learning can depend on the type of information encountered, how often the information is encountered, what kind of person you are, and what kind of people you encounter [10]. In this paper, we test two processes about how the identities of the learner and the person they learn from affect social learning. One process suggests that people choose who to learn from based on *prestige*, a person's reputation of success or knowledge [13]. A second process suggests that social learning is based on *homophily*- the idea people are more likely to learn from people who are similar to them (e.g., same-age, same-family, or same-gender) [14]. Although these processes are both well-supported individually, there is limited evidence of how they concurrently relate to social learning within the same network [10, 12].

One of the most fundamental knowledge domains for human survival is ecological knowledge (cultural adaptations to ecological environments) because in pre-industrial societies this knowledge can make the difference between life and death [15, 16]. For example, in Indigenous subsistence societies hunting knowledge is associated with personal health [17] and medicinal plant knowledge is linked with offspring health [18, 19] and social influence [20]. Medicinal plants remain relevant to contemporary survival, since plants are still the central component of health care for ~80% of the world's population, are used to make ~25% of pharmaceutical drugs [21], and provide a global market worth over $84 billion [22]. Medicinal plants also support biocultural diversity, since at least 18,000 plant species are used for medicine- more species than any other type of ecological knowledge, including food and construction [22]. Thus, understanding the distribution and transmission of ecological knowledge has critical implications for conservation, health, and economy, and exploring fundamental theories about social learning. However, no studies have tested whether hypotheses derived from both homophily and prestige can predict the configuration and learning patterns of medicinal plant knowledge networks [23, 24].

Testing how homophily and prestige affect sharing of medicinal plant knowledge provides critical insight for conservation of biological and cultural diversity. Both biological diversity and ecological knowledge are essential to provide the benefits humans get from ecosystems (i.e., ecosystem services), such as food production, climate change regulation, recreation, and water purification [25]. Ecological knowledge also contributes to biological conservation and restoration [26–28], while biological diversity and conservation contribute to ecological

knowledge [29, 30]. Because ecological knowledge, biodiversity, cultural diversity, and conservation are interconnected, 80% of the world's biodiversity is managed by Indigenous Peoples [31, 32]. However, worldwide biological and cultural changes [33, 34] are a threat to ecological knowledge, especially medicinal plant knowledge [35]. Therefore, understanding how ecological knowledge is shared, especially by Indigenous Peoples, will inform management of both ecological knowledge and biodiversity around the world.

Following an approach combining semi-structured interviews, participant observation, checklist interviews, and social network analysis, we assess three hypotheses about the social learning processes that shape medicinal plant knowledge networks: prestige predicts network configuration, prestige predicts learning patterns, and homophily predicts learning patterns. These three hypotheses were further divided into subhypotheses to investigate the effects they predict on network patterns (Fig 1 and Table 1). To test these hypotheses, we use data from a network of medicinal plant knowledge sharing among residents of four subsistence villages in Solomon Islands that primarily share the same kinship lineage.

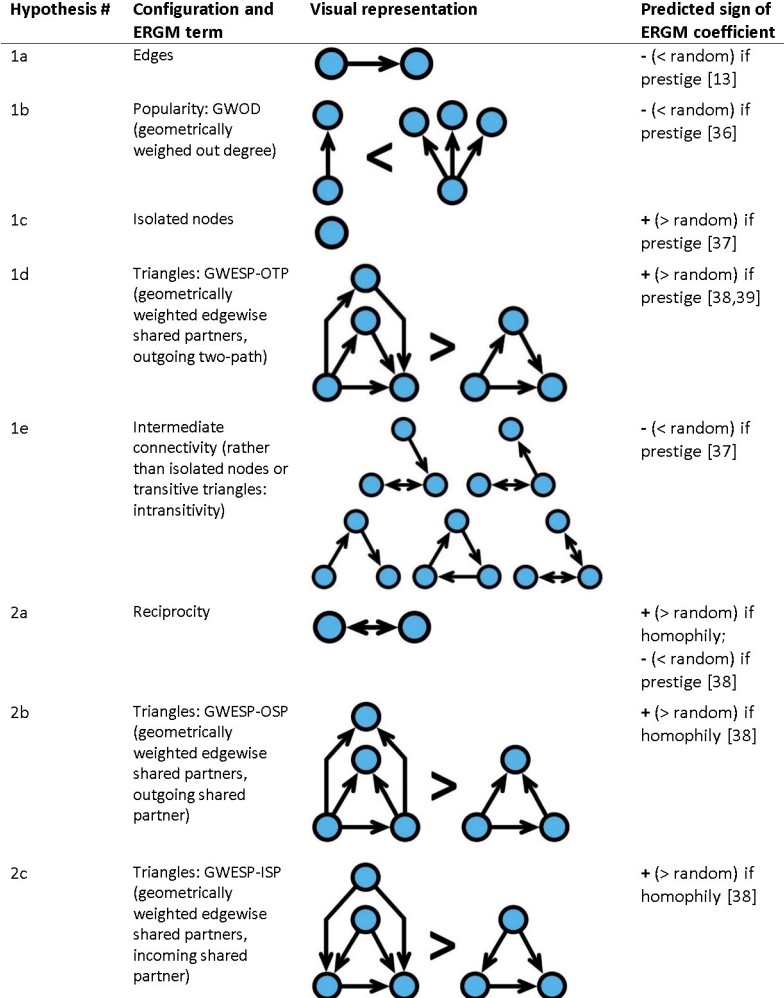

**Fig 1. Hypothesized predictors of how prestige and homophily-based learning patterns affect medicinal plant knowledge network configuration.** Network configuration descriptions include terms used in exponential random graph modeling (ERGM) [13, 36–39].

**Table 1. Hypothesized predictors of how prestige and homophily affect learning patterns in medicinal plant knowledge networks.**

| Hypothesis # | Predictor of popularity | Predicted sign of ERGM coefficient | Explanation |
|---|---|---|---|
| 3a | Gender (man) | + [40] | Prestige: men tend to have more knowledge than women in the region of Melanesia due to cultural power dynamics [41,42] |
| 3b | Age | + [19] | Prestige: older individuals are more likely to be asked for knowledge [10,15] |
| 3c | Medicinal knowledge | + [40,43] | Prestige: skillful individuals are more likely to be asked for knowledge [10,15] |
| Hypothesis # | Predictor of sameness | Predicted sign of ERGM coefficient | Explanation |
| 4a | Descendant (children and grandchildren) | + [19] | Homophily: genetically similar (closely related) people are more likely to share knowledge [19,44] |
| 4b | Descendant of main kinship group (distant kinship) | + [40,45] | Homophily: individuals who are patrilineal descendants of the single kinship group that is most common across all four villages are more likely to share knowledge because they are related [19,44] |
| 4c | House distance (general kinship) | - [39] | Homophily: in this region most marriages are patrilocal and nearby houses tend to be related by descent or marriage [46], and are therefore more likely to share knowledge [19,44] |
| 4d | Spouse | + [19] | Homophily: spouses are more likely to share knowledge [19] |
| 4e | Village | + [39] | Homophily: individuals from the same village are all very familiar [46], and may be more likely to also share knowledge [39,47] |
| 4f | Gender | + [19] | Homophily: in this region men and women are historically very socially segregated, unless they are related [46], so knowledge tends to be shared between people of the same gender [19] |
| 4g | Age | -* if homophily [19]; | Homophily: peers are more likely to share knowledge [19] |
|  |  | + if prestige [15] | Prestige: knowledge is more likely to be shared from older to younger [10,15] |
| 4h | Medicinal knowledge | -* [43] | Control variable to distinguish between stated and revealed knowledge sharing [48]. Individuals who say they share knowledge should actually have similar knowledge [43,49]. |

*in ERGM, similarity of continuous variables is indicated by a negative coefficient

## Theoretical Framework: Linking social learning processes to network structures

Here, we explore how homophily and prestige relate to social learning of ecological knowledge, and how these processes affect which nodes of a network are connected and specific patterns of network connections. By explaining the implications of these social learning processes, we develop hypotheses that are tested using structural signatures in a network of medicinal plant knowledge sharing. To test our hypotheses, we compare the frequency of relevant network statistics in our observed data with the distribution of statistics to random graphs of the same structure as the observed data, called ERGMs (exponential random graph models), which control for simultaneous effects of multiple social processes [37]. It is critical to recognize that ERGMs do not directly observe the exact mechanisms of social learning, such as the personal motivation for requesting medicinal knowledge from specific individuals. However, if prestige and homophily affect knowledge sharing, specific network configurations will be more common or rare than in a random network [23]. Therefore, presence and frequency of certain structural components in a network allows us to infer which social learning and cultural adaptation processes are influencing that network [50].

**Hypothesis 1: Prestige predicts knowledge sharing network configuration.** By its very nature, prestige is exclusive, typically shared selectively with kin or offspring to increase fitness [13, 19]. This is particularly true for medicinal knowledge in Solomon Islands, the area where this study takes place [51]. As a result of this restricted sharing, we predict that there will be fewer knowledge sharing relationships (called *edges*) in the observed network than in a random network of the same size (Hypothesis 1a, Fig 1).

A related concept, called centralization or preferential attachment, suggests that nodes who are connected to one other node are more likely to be make new connections with other nodes [52]. Therefore, the distribution of edges is not even among nodes [39]. In the ERGM context, we test for network popularity with a geometrically weighted out degree (GWOD) term, which measures whether people who share knowledge with at least one person are more likely to share knowledge with more than one person [37]. Thus, we predict that fewer nodes will have high popularity ($\geq 2$ edges) and more nodes will have low popularity ($\leq 1$ edges) in the observed network than in a random network (Hypothesis 1b, Fig 1). Because sharing is restricted, certain nodes may not meet the selective criteria for sharing knowledge. Therefore, we also predict that in our network more nodes will have no edges connecting them to another node than in a random network (Hypothesis 1c, Fig 1).

Another well-established network property is closure, the tendency for nodes to form edges with their network partners' partners [37, 39]. This redundancy helps to improve the accuracy of social learning and ensure the resiliency of cultural adaptation if nodes are removed from the network [53, 54]. In the context of social learning, prestige predicts more triangles than a random network because people are likely to learn from their teacher's teacher, with an increased chance for every teacher that the node and the teacher's teacher have in common (Hypothesis 1d, Fig 1) [38]. In ERGMs, we test for these types of triangles with a geometrically weighted edgewise shared partners, outgoing two-path (GWESP-OTP) term [37, 38].

Combining the ideas of centralization and closure suggests that medicinal knowledge sharing tends to be absent (leaving many nodes isolated or poorly connected) or redundant (with clustered edges that connect a few nodes). Therefore, we predict that the observed network will have less intermediate connectivity than a random graph (Hypothesis 1e, Fig 1).

**Hypothesis 2: Homophily predicts knowledge sharing network configuration.** Homophily predicts that you are more likely to learn from someone who is like you in some way. As a result of that similarity, so the person you learn from is also more likely to learn from you. Therefore, we predict that there will be more reciprocal relationships in the observed network than in a random network of the same size (Hypothesis 2a, Fig 1). If prestige drives learning patterns, the prediction for reciprocity is inverted because prestigious people don't need to learn from less prestigious people (Hypothesis 2a, Fig 1). Closure has already been mentioned above, but here we discuss two alternate triangle structures. Homophily predicts a higher chance of node "A" learning from node "B" if they both learned from person "C" or if person "C" learned from both of them, because person "C" must be like both of them, which means "A" and "B" are like each other [37, 38]. In ERGMs, we test for these types of triangles with geometrically weighted edgewise shared partner terms for outgoing shared partner (GWESP-OSP) and incoming shared partner (GWESP-ISP) (Hypotheses 2b-c, Fig 1).

**Hypothesis 3: Prestige predicts learning patterns in knowledge sharing networks.** The primary sources of prestige for learning patterns are knowledge, age, and gender, particularly older and more successful people [13, 15]. If prestige for any of these variables drives social learning, we expect to find patterns of popularity (the number of people who say they learned from a given person) in our observed network (Hypotheses 3a-c, Table 1). In the cultural context of Solomon Islands, men and women have distinct gender roles and restrictions, and men tend to have more political power and ancestral knowledge than women, so men will tend to have more prestige [46, 51].

**Hypothesis 4: Homophily predicts learning patterns in knowledge sharing networks.** Homophily encompasses many forms of similarity, including education, religion, and occupation, and affects many forms of social connection, such as friendship, membership, and sharing possessions [14]. However, in the cultural context of Solomon Islands, the most likely sources of homophily are house proximity, marriage, village of residence, descent group,

gender, and age [46, 55]. If homophily of any of these variables drives learning patterns, we expect to find similarity between people who share knowledge in our observed network (Hypotheses 4a-g, Table 1). If prestige drives learning patterns, the prediction for one variable, age, is inverted (Hypothesis 4g, Table 1). We also predict another form of similarity, medicinal knowledge, not because of homophily but as a result of knowledge sharing (Hypothesis 4h, Table 1). This variable is included to control for potential differences between stated and revealed social learning relationships [48]. Although participants could potentially state that they learned from an influential or important person, revealed knowledge sharing is supported by similarity in medicinal plant knowledge [43, 49].

## Materials and methods

### Fieldwork site

Our study populations are four Indigenous forager-horticulturist villages in Solomon Islands. Solomon Islands is in the South Pacific, northeast of Australia (Fig 2). For 13 months (Dec. 2014 –Sept. 2017), the first author lived in four villages in Malaita, Solomon Islands (Fig 2). This region was selected because of the continued reliance on herbal medicines [46]. Research villages were selected because most residents share the same kinship lineage, which means that they are blood descendants from a single ancestor. Residents of two villages (Kolofi and Binaoli) 1.5 kilometers apart are descended from families who left their homeland region ~200 years ago because of inter-family conflict, and primarily speak Baelelea language. Residents of the other of two villages (Irobulu and Lagoe) 1 kilometer apart are descended from the families who remained in their homeland region and primarily speak Baegu language. Residents in each pair of villages are aware of their relationships with residents in the other pair, but rarely visit each other because they are 26 kilometers apart. Five native languages are commonly spoken in north Malaita, as well as Solomon Islands Pijin, and most residents can understand several languages [46], so residents of all four villages are able to communicate. Other than the difference in primary language, the cultural systems of the villages are very similar.

All four villages are in tropical wet forests with similar plant species richness [30]. The main activities for inhabitants of these villages are slash-and-burn subsistence agriculture, building or repairing thatch houses, and Christian church activities [46, 56, 57]. In this region medicinal plants are widely used to treat medical diseases and illnesses caused by supernatural or spiritual forces [46, 56]. Each village is 2–4 kilometers from the nearest rural health clinic, and rural clinics often have no medicines or staff [30, 58]. Additionally, Kolofi is the only village adjacent to a gravel road- residents in the other three villages must walk 1.5–4 kilometers to access a road, where transport to larger hospitals is possible. Because medical centers have limited supplies, irregular hours, and are hard to reach, residents of the study villages largely depend on herbal remedies. Because medicinal knowledge is highly valued in this region, it is often hidden from strangers and only shared with relatives or close friends. Herbal preparations may be shared with sick acquaintances if asked, but some form of return payment is often expected [46, 56].

### Ethnographic and medicinal plant knowledge sampling

Before beginning this project, a permit from the Solomon Islands Ministry of Education and Human Resources Development (research permit #2015/024), community support letters, and University of Hawaiʻi ethics approval (IRB #22589) were obtained. Interviews included an oral record of free and informed prior consent and were conducted in three local languages: Baeggu, Baelelea, and Solomon Islands Pijin. In all four research villages every consenting resident over age 18 (303 participants) was interviewed twice (S1 Table, S1 File). Interview 1 was

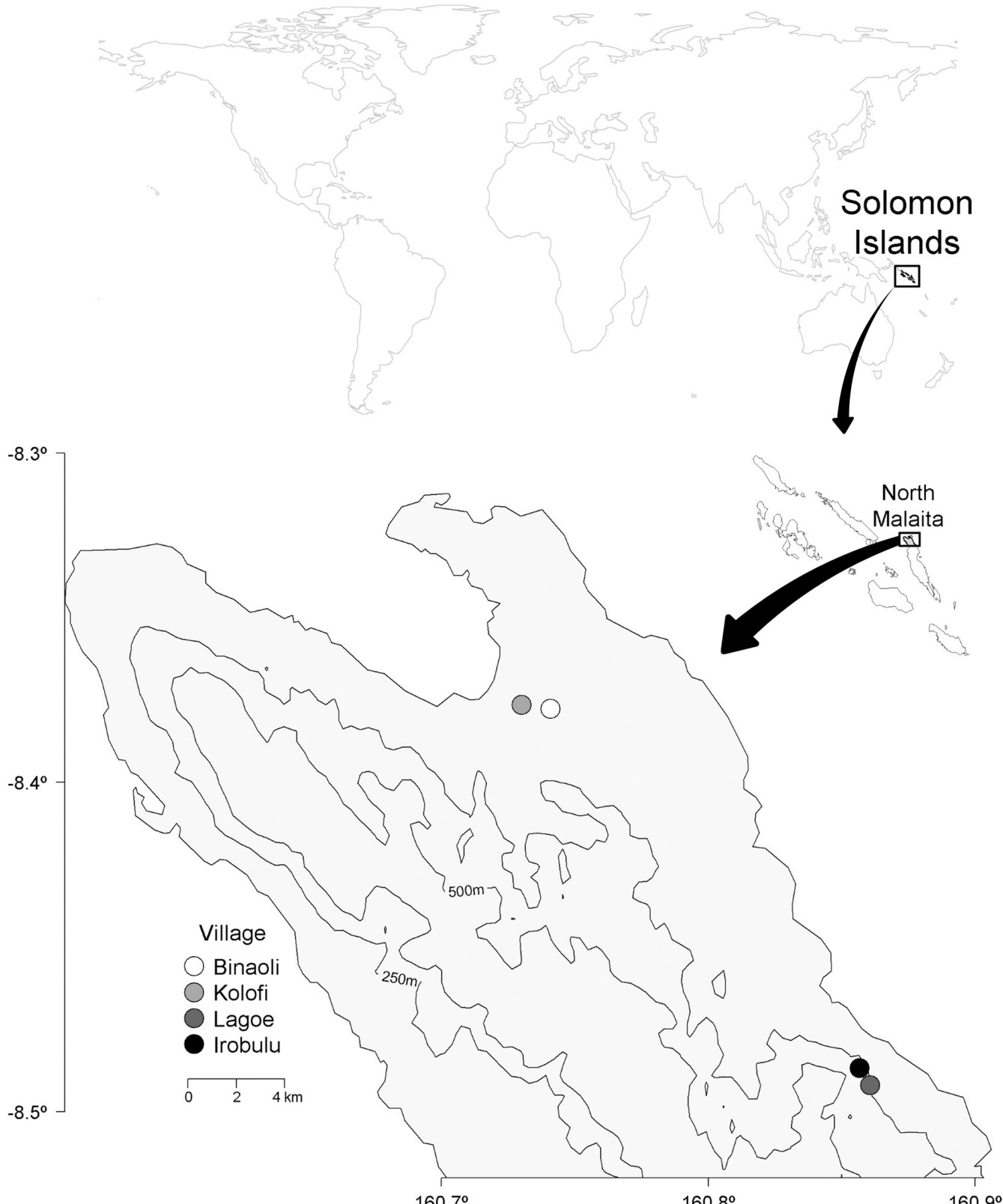

**Fig 2. Map of four research villages, north Malaita Island, Solomon Islands.** Contour lines represent 250m and 500m elevation. Locator map in upper right shows region of study site within the nation of Solomon Islands. Map created by the authors in R with coastline data from Natural Earth [59] and elevation data from Viewfinder Panoramas [60].

sometimes conducted with a local assistant who helped reassure nervous participants and assist translation. Interview 2 was conducted individually whenever possible to avoid influences from third parties and to assure that the data supplied by the resident were as direct and reliable as possible [61].

Interview 1 collected sociocultural and demographic data, including GPS coordinates of the interviewees house (S1 Table), and a list of the plants the interviewee uses as medicine. Because free-listing can exclude important information [62], this technique was corroborated and supplemented by participant observation and walk-in-the-woods guided tours [61]. Participant observation, which followed the residents' daily activities without interfering with them, clarified information collected in interviews. Walk-in-the-woods guided tours were conducted with residents who volunteered to identify species mentioned in the free lists. Every free-listed plant was photographed and sampled for voucher specimens (S1 Appendix). Voucher specimens were deposited in herbaria of the Smithsonian Institution (US), the University of Hawai'i (HAW), and Solomon Islands (BSIP). Final identification was accomplished by crosschecking published sources that include local names and taxonomic keys [56, 63, 64]. Species synonymies were resolved following the Plant List database v. 1.1 (http://www.theplantlist.org/, accessed 04/29/2018).

In the second interview, for each village, photographs of every free-listed plant were compiled and shown to participants on a laptop in a checklist interview, to provide identical stimulus for all participants and facilitate quality and quantity of information collected [61, 65, 66]. Because photographs provide less sensory context than *in situ* specimens, each photograph was accompanied by an oral description that included local names for the plant and ecological information. In the checklist interview, each resident identified which plants from the photo set they use medicinally, and which illnesses or symptoms each plant is used to treat. Interview 2 included 177–249 species (S1 Table) and took 20–120 minutes to complete, depending on the knowledge of the interviewee. To reduce interview fatigue, participants were offered betel nut during the interview, a common activity for socializing [46]. Plant uses were categorized based on the International Classification of Primary Care [67], with an additional category for animal medicine [68] (S2 Appendix). For example, one plant species cited to treat one kind of eye illness and two kinds of digestive illnesses was analyzed as one species, two illnesses, and three uses.

## Medicinal plant knowledge

Five types of medicinal plant knowledge were evaluated: the number of uses cited, species cited, and illnesses cited, as well as the use uniqueness and species uniqueness. The number of illnesses cited, species cited, and uses cited are commonly used in ethnobotanical studies to quantify the amount of knowledge [61]. To estimate use and species uniqueness, we used local contribution to beta diversity, which measures the variation in people's knowledge by comparing their knowledge to all knowledge in the entire village. Scores range from 0–1; higher scores indicate that the person's knowledge is unusual and unique, and lower scores indicate that the person's knowledge is common and widely shared [69]. For each person, contributions to beta diversity of species cited and uses cited were calculated in *adespatial* R package [70] using Euclidean distance to account for meaningful zeros in binary data (species cited) and count data (uses cited). Sampling adequacy of medicinal plant species was assessed using the *iNEXT* package to calculate rarefaction curves and sample coverage estimates (S1 Fig) [71].

## Medicinal plant knowledge sharing network construction

We characterized the network of medicinal plant knowledge sharing using a name generating procedure. At the end of the second interview, each participant was asked how they learned

about medicinal plants. The participant was also asked to cite all the people who ever taught them about medicinal plants in their life, and explain their relationship (e.g., friend, aunt, parent). We analyzed the closed medicinal knowledge sharing network among the 303 participants to characterize the influence of homophily and prestige. The matrix of incoming knowledge sharing was coded 1, irrespective of the number of exchange events involved, or 0 if no exchange was reported. Exchange edges were represented as a 303 × 303 directed matrix represented by the following relationship: "person *i* learned medicinal plant knowledge from person *j*." Open and closed networks were visualized using the *igraph* R package (Fig 3, S2 Fig) [72]. Summary statistics were calculated for each village (transitivity, density, diameter) and person (in degree, out degree) within the network using the *igraph* R package [72].

## ERGM analysis

The goal of ERGMs is to predict the network structure by using network metrics which represent social learning processes that are hypothesized to generate the network. Because ERGMs calculate the conditional log-odds of individual edges, coefficients are interpreted in the same way as a standard logistic regression; positive and negative parameters represent processes that increase or decrease the probability of an edge, respectively. To calculate the conditional probability of an edge given a combination of processes, parameter estimates for the estimates are summed [37].

Before ERGM analysis, we visualized correlations between nodal variables by calculating Pearson correlation coefficients (S3 Fig) with the *corrplot* package [73]. Because all five medicinal knowledge variables are highly correlated, each type of knowledge was analyzed separately. ERGMs were calculated using the *ergm* command in R [74, 75]. Because descent, marriage and house distance depend on data from the entire network, they were calculated as undirected 303 × 303 matrices. For descent, children, grandchildren, parents, and grandparents were coded 1 and all other relationships were coded 0. For marriage, spouses were coded 1 and non-spouses were coded 0. To measure the distance between houses, we calculated a matrix of the geodesic distance between each house using the *geosphere* r package [76]. To reduce influence of positively skewed values, distance was log (x + 1)-transformed [77]. Remaining demographic data (e.g., age, gender) were added to the entire closed network as nodal attributes.

ERGMs were constructed by adding terms following the methods of Goodreau et al. [78], starting with the structural covariates (M1), including the two geometrically-weighted structural variables (GWOD and GWESP). For GWOD we specified a fixed decay-parameter value of $\theta S = 0.5$ to apply equal popularity effects to nodes with high and low out degree [37]. For the three GWESP terms we specified a fixed decay-parameter value of $\theta T = 0.01$ to represent a higher chance of node "A" learning from node "B" if they both have social learning relationships with person "C", with a very slightly increased chance for every person that "A" and "B" both have social learning relationships with (Fig 1 H1d, H2b-c) [37]. One additional model (M2) was constructed by adding all edge and node variables except medicinal plant knowledge to M1. Five additional models (M3-7) were constructed by one of the five measurements of medicinal plant knowledge to M2. After all seven ERGMs were complete, Variance Inflation Factor (VIF) values were calculated using the *vif.ergm* command for each model term [79]. Model terms with VIF values greater than 10 ("use uniqueness: same" and "species uniqueness: same") were removed to avoid multicollinearity [79]. To confirm that adding terms improved model fit we used Akaike Information Criterion (AIC) to test if more complex models had lower AIC than simpler models (M3-7<M2<M1), which indicates a better fit [80]. ERGM goodness-of-fit (S4–S10 Figs) and Markov chain Monte Carlo (MCMC) diagnostics (S11–S17 Figs) were used to evaluate model performance. To test alternative explanations for non-

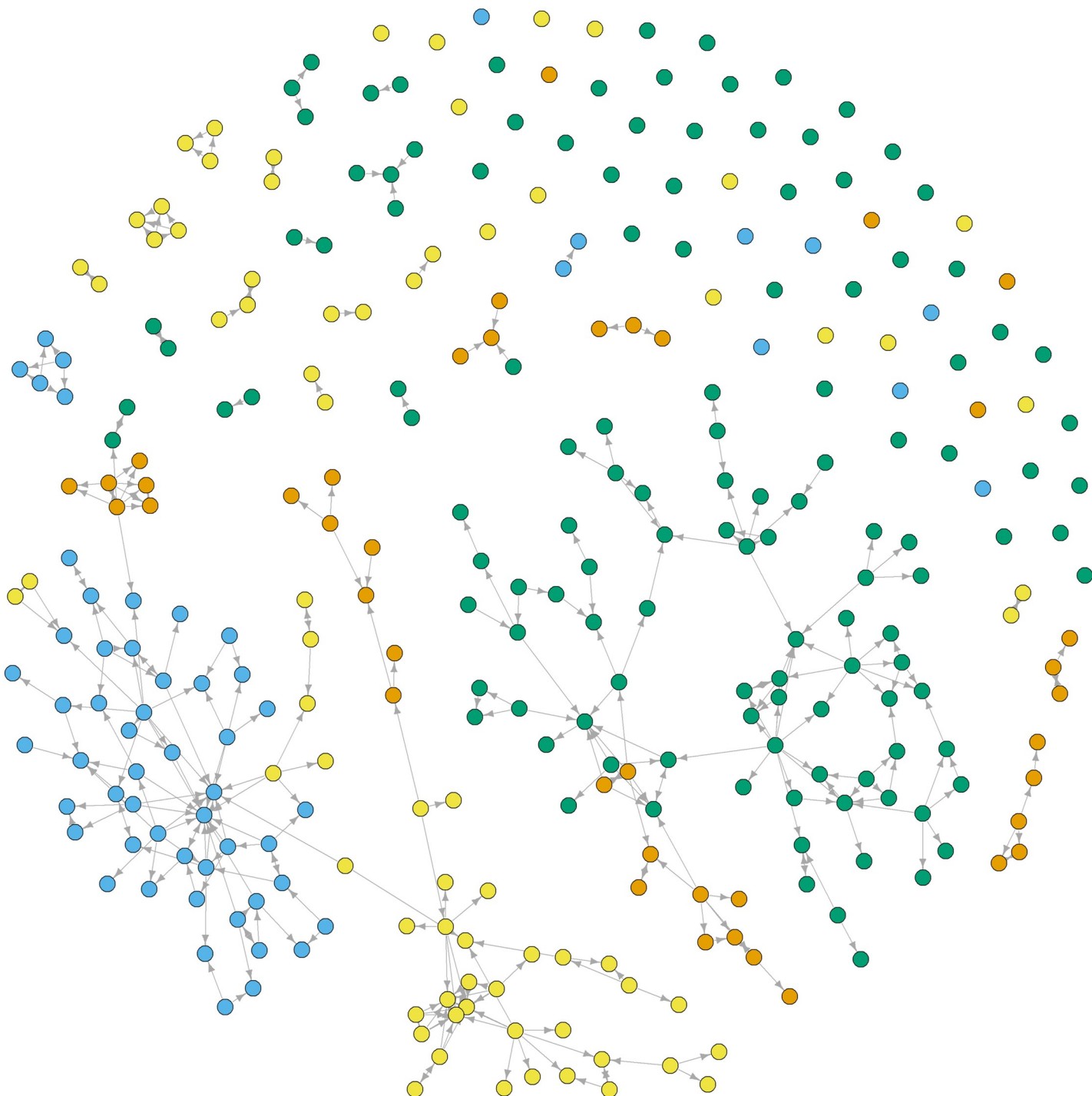

**Fig 3. Fruchterman-Reingold representation of closed medicinal plant knowledge network among the 303 participants.** Dots representing participants are colored according to village (Binaoli = orange, Kolofi = green, Irobulu = blue, Lagoe = yellow). Arrows connecting dots represent flow of knowledge shared from one person to another.

significant results we visualized the probability of edges forming between all nodes in the network using the *edgeprob* command [81]. All analyses were performed in R version 3.6.1 [82].

## Results

### Descriptive results

Across all villages a total of 317 plant species from 96 families were cited by participants as medicinal (S1 Table, S1 Appendix). Sample coverage estimates for medicinal plant species known in each village all exceeded 99% (S1 Fig). The closed medicinal plant knowledge sharing network is shown in Fig 3. Kolofi is the largest, least dense, and least centralized village; it also has the largest number of isolated nodes and has the lowest medicinal knowledge (S1 Table). Binaoli is the smallest village and has the smallest number of isolated nodes. Binaoli and Lagoe have the most clustering. Irobulu has the highest density, the least clustering, and the largest diameter. Irobulu and Lagoe have the highest in degree and centralization, while Irobulu and Binaoli have the highest out degree. AIC confirmed that model complexity improved model fit (M3-7<M2<M1). Model validation indicated no problems (S4–S10 Figs).

### Hypothesis 1: Prestige predicts knowledge sharing network configuration

The results of medicinal plant knowledge sharing network ERGMs are shown in Table 2. All configuration variables are significant in all models, except for popularity in M5 (# illnesses known). Consistent with H1a, edge estimates were strongly negative, indicating restricted sharing of medicinal plant knowledge. Edge estimates are also consistently more negative in the models that include edge and node variables (M2-7). For example, in M1 nodes have a 1% chance of being connected (compared to 50% chance in a random network). In M2-7, nodes have a 0.05–0.01% chance of being connected, which suggests that edge and node effects in M2-7 explain more detail of how nodes connect, and that nodes are highly unlikely to connect without the influence of homophily or prestige. Consistent with H1b, estimates for popularity are significantly negative in M1 and M2, indicating that people are more likely to learn about medicinal species and uses from someone who at least one other person has also learned from, but that medicinal knowledge is more important than popularity which choosing who to learn from.

Consistent with H1c, estimates for isolated nodes are consistently positive, indicating that more people in the observed network are not socially learning medicinal plant knowledge than would be expected in a random network. These estimates are roughly the same across M1-7 (70–75% change of a node being isolated), which suggests that the likelihood a person not sharing or learning medicinal knowledge in these villages remains the same, regardless of edge and node effects. In contrast to H1d, estimates for OTP triangles are consistently negative, but not significant. Consistent with H1e, estimates for intermediate connectivity are consistently negative, indicating that medicinal plant knowledge sharing tends to be absent (leaving many nodes isolated or poorly connected) or redundant (with clustered edges that connect a few nodes). These estimates are similar across M1-7 (31–35% change of intermediate connections between nodes), which suggests that the likelihood of intermediate connective remains the same, regardless of edge and node effects.

### Hypothesis 2: Homophily predicts knowledge sharing network configuration

Consistent with H2a's homophily prediction, estimates for reciprocity are significantly positive in M1; however, significant negative estimates in M2-M7 support H2a's prestige prediction (Table 2). The change in sign indicates that the reciprocity in this network is explained by homophily of people who are direct descendants, spouses, or from the same village. Reciprocity estimates are similarly negative across M2-7 (26–33% chance), which suggests that after

**Table 2. Results of Exponential Random Graph Models (ERGMs).**

| | M1: configuration variables | M2: M1 + edge & node variables | M3: M2 + # uses known | M4: M2 + # species known | M5: M2 + # illnesses known | M6: M2 + use uniqueness | M7: M2 + species uniqueness |
|---|---|---|---|---|---|---|---|
| **Configuration: Prestige** | | | | | | | |
| Edges | -4.09 (0.10) *** | -7.67 (0.46) *** | -8.40 (0.53) *** | -8.62 (0.52) *** | -9.57 (0.65) *** | -8.15 (0.49) *** | -8.40 (0.50) *** |
| Popularity (GWOD, $\theta S = 0.5$) | -1.29 (0.23) *** | -0.64 (0.29) * | -0.46 (0.30) | -0.46 (0.31) | -0.33 (0.32) | -0.50 (0.32) | -0.48 (0.31) |
| Isolated nodes | 0.86 (0.22) *** | 1.05 (0.24) *** | 1.04 (0.25) *** | 1.00 (0.24) *** | 1.08 (0.25) *** | 1.06 (0.24) *** | 1.11 (0.25) *** |
| Triangles (GWESP-OTP, $\theta T = 0.01$) | -0.85 (0.66) | -0.47 (0.50) | -0.35 (0.45) | -0.46 (0.45) | -0.37 (0.44) | -0.35 (0.43) | -0.40 (0.42) |
| Intermediate connectivity (intransitivity) | -0.78 (0.07) *** | -0.65 (0.08) *** | -0.62 (0.08) *** | -0.63 (0.08) *** | -0.65 (0.08) *** | -0.65 (0.08) *** | -0.64 (0.08) *** |
| **Configuration: Homophily** | | | | | | | |
| Reciprocity | 3.58 (0.40) *** | -0.81 (0.37) * | -0.83 (0.37) * | -0.76 (0.39) * | -1.03 (0.39) ** | -0.80 (0.39) * | -0.73 (0.37) * |
| Triangles (GWESP-OSP, $\theta T = 0.01$) | -0.24 (0.28) | -0.00 (0.31) | -0.06 (0.29) | 0.10 (0.31) | 0.00 (0.27) | 0.02 (0.30) | -0.06 (0.29) |
| Triangles (GWESP-ISP, $\theta T = 0.01$) | 2.72 (0.62) *** | 1.08 (0.47) * | 1.02 (0.46) * | 0.99 (0.48) * | 0.97 (0.44) * | 0.93 (0.42) * | 1.01 (0.43) * |
| **Node variables: Prestige** | | | | | | | |
| Gender (man): popularity | | 0.46 (0.13) *** | 0.36 (0.14) * | 0.40 (0.14) ** | 0.52 (0.15) *** | 0.42 (0.14) ** | 0.41 (0.14) ** |
| Age: popularity | | 0.04 (0.01) *** | 0.04 (0.01) *** | 0.04 (0.01) *** | 0.04 (0.01) *** | 0.04 (0.01) *** | 0.04 (0.01) *** |
| # uses known: popularity | | | 0.02 (0.00) *** | | | | |
| # species known: popularity | | | | 0.02 (0.00) *** | | | |
| # illnesses known: popularity | | | | | 0.16 (0.03) *** | | |
| use uniqueness: popularity | | | | | | 27.05 (3.92) *** | |
| species uniqueness: popularity | | | | | | | 37.92 (5.18) *** |
| **Node variables: Homophily** | | | | | | | |
| Direct descendant | | 3.67 (0.21) *** | 3.65 (0.21) *** | 3.67 (0.21) *** | 3.66 (0.20) *** | 3.63 (0.22) *** | 3.70 (0.21) *** |
| Descendant of main kinship group | | 0.31 (0.16) * | 0.27 (0.16) | 0.29 (0.17) | 0.33 (0.16) * | 0.29 (0.16) | 0.31 (0.16) |
| House distance | | -1.45 (0.23) *** | -1.43 (0.27) *** | -1.37 (0.25) *** | -1.54 (0.22) *** | -1.56 (0.24) *** | -1.45 (0.25) *** |
| Spouse | | 4.57 (0.33) *** | 4.53 (0.36) *** | 4.47 (0.35) *** | 4.59 (0.35) *** | 4.51 (0.33) *** | 4.49 (0.36) *** |
| Village: same | | 2.86 (0.26) *** | 2.89 (0.27) *** | 2.90 (0.26) *** | 2.90 (0.26) *** | 2.98 (0.27) *** | 3.02 (0.26) *** |
| Gender: same | | 0.64 (0.16) *** | 0.63 (0.17) *** | 0.64 (0.16) *** | 0.67 (0.16) *** | 0.67 (0.16) *** | 0.66 (0.16) *** |
| Age: same | | -0.01 (0.01) | -0.00 (0.01) | -0.00 (0.01) | -0.00 (0.01) | -0.00 (0.01) | -0.01 (0.01) |
| # uses known: same | | | -0.01 (0.00) ** | | | | |
| # species known: same | | | | -0.02 (0.00) *** | | | |
| # illnesses known: same | | | | | -0.11 (0.03) *** | | |
| AIC | 3816.55 | 2175.79 | 2120.21 | 2119.1 | 2116.8 | 2138.86 | 2136.46 |
| Log Likelihood | -1900.27 | -1070.9 | -1041.1 | -1040.55 | -1039.4 | -1051.43 | -1050.23 |

Parameter estimates are expressed in log-odds with their standard error (SE) in parentheses.

*P < 0.05

**P < 0.01

***P < 0.001. GWOD = geometrically weighted out-degree distribution; GWESP = geometrically weighted edgewise shared partner; OTP = outgoing two-path; OSP = outgoing shared partner; ISP = incoming two-path.

accounting for node homophily, the lack of reciprocity supports prestige. In contrast to H2b, estimates for OSP triangles are close to zero and not significant. Consistent with H2c, estimates for ISP triangles are significantly positive, indicating that more people are more likely to learn from each other if they both learn from the same person, particularly if they both learn from the same two or more people. These estimates are higher in M1 (94% chance) than in M2-7 (72–75% change), which suggests that people who are direct descendants, spouses, or from the same village also tend to have the same teachers.

## Hypothesis 3: Prestige predicts learning patterns in knowledge sharing networks

Both of the non-medicinal prestige variables are significant in all models (Table 2). Estimates are similar across M2-7, which suggests that the likelihood of learning based on prestige remains the same, regardless of whether they know about medicinal plants. Consistent with H3a, gender-popularity estimates were consistently positive, indicating that people are more likely to learn about medicinal plants from men (59–63% chance). Consistent with H3b, age-popularity estimates were consistently positive, indicating that people are more likely to learn about medicinal plants from older people (51% chance). Consistent with H3c, knowledge-popularity estimates were consistently positive, indicating that people are more likely to learn about medicinal plants from more knowledgeable people (51–100% chance). These estimates are stronger in M3-5 (51–54% chance) than in M6-7 (100% chance), which suggests that "experts" in medicinal plants do not just know more than other people, they know uses and species that are not known by others in their social group. This result also supports the fundamental assumption that prestige is associated with skill or success [13].

## Hypothesis 4: Homophily predicts learning patterns in knowledge sharing networks

Five of the seven non-medicinal homophily variables are significant in all models (Table 2). Estimates are similar across M2-7, which suggests that the likelihood of learning based on homophily remains the same, regardless of how much they know about medicinal plants. Consistent with H4a, direct descendant estimates were consistently and strongly positive, indicating that medicinal plant knowledge is more likely to be taught by parents and grandparents to their descendants (97% chance).

Consistent with H4b, same-kinship group estimates were positive in M2-7, but not significant in M2 and M 5 (58%), indicating that descendants of the main kinship descent group are more likely to learn from other kinship group members about illnesses, but not about medicinal species or uses. In M2-7 the probability of an edge forming between two people who are not both descendants of the main kinship group is higher for man-woman pairs than for same-gender pairs, but same-gender pairs are consistently more likely to be connected by an edge if they are both members of the main kinship group (Fig 4), indicating that nonsignificance of kinship descent homophily may be related to cultural gender roles.

Consistent with H4c, house distance estimates were consistently and strongly negative, indicating that medicinal plant knowledge is more likely to be taught and learned by people who live close to each other. Consistent with H4d, spouse estimates were consistently and strongly positive, indicating that medicinal plant knowledge is more likely to be taught and learned by spouses (99% chance). Consistent with H4e, same-village estimates were consistently and strongly positive, indicating that medicinal plant knowledge is more likely to be taught and learned by residents of the same village (95% chance).

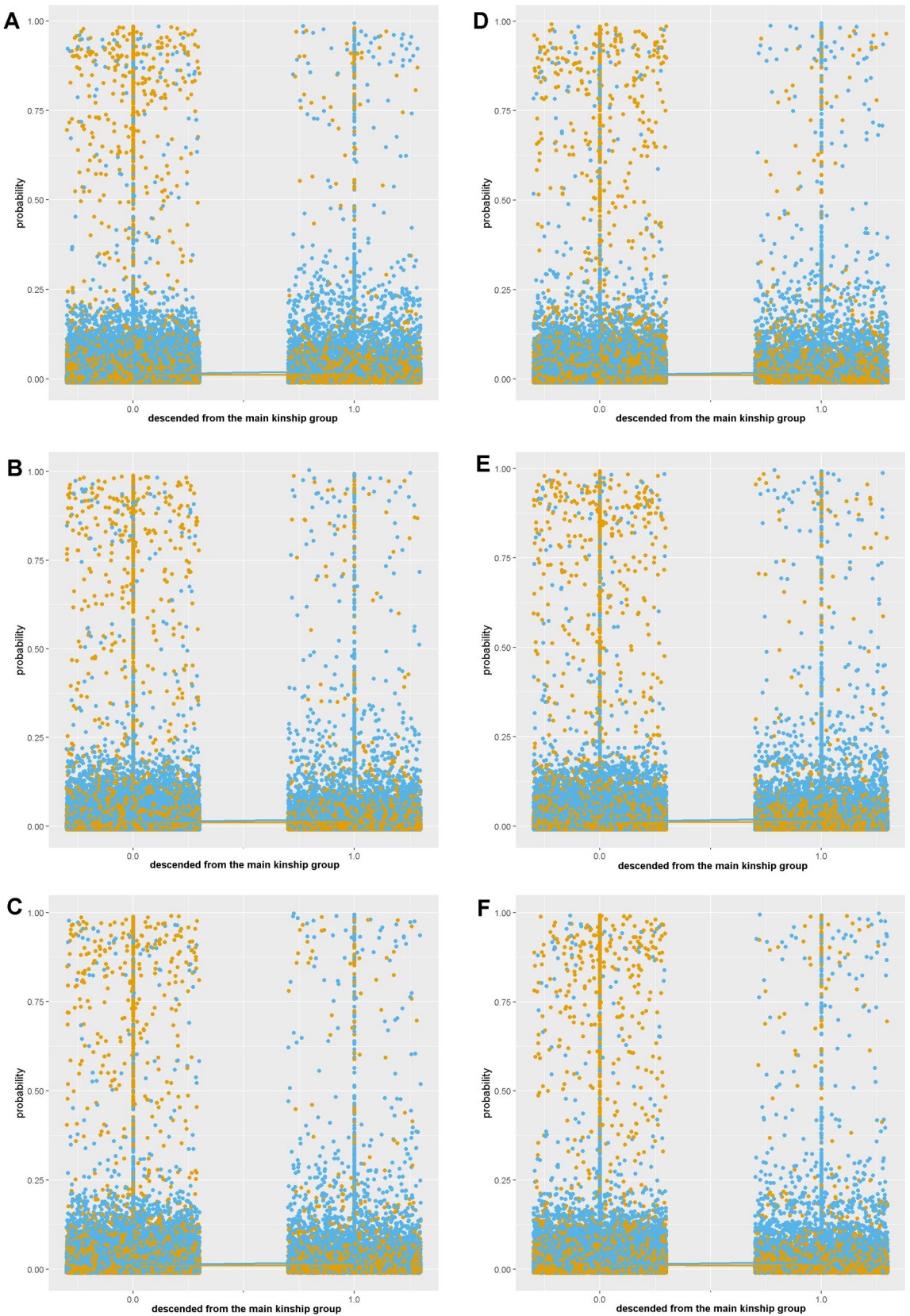

**Fig 4. Probability of an edge forming between each possible pairing of the 303 people in the network who are (1.0) or are not (0.0) both from the main kinship group.** Dots representing the probability of each possible edge calculated from exponential random graph models are orange for man-woman pairs and blue for same-gender pairs. Key: A = M2, B = M3, C = M4, D = M5, E = M6, F = M7.

Consistent with H4f, same-gender estimates were consistently positive, indicating that medicinal plant knowledge is more likely to be taught and learned by people who share the same gender (65–66% chance). To address H4g, same-age estimates were all very slightly negative and not significant, indicating no pattern of whether medicinal plant knowledge is taught and learned by people who are or are not the same age (50% chance).

In contrast to H4g, we find no support for age homophily. In M2-7 the probability of an edge forming between two people whose age difference is <10 years is higher for man-woman pairs than for same-gender pairs (Fig 5); however, this is not an indication of age homophily, but spousal homophily. Culturally, relationships between men and women are discouraged, especially if they are close in age, unless they are spouses. Therefore, increased sharing of knowledge between spouses (who are close in age) may mask a slight positive effect of age prestige. Consistent with H4h, same-knowledge estimates were consistently negative, indicating that people tend to have similar medicinal plant knowledge to people they learned from (50–53% chance). This result confirms that people who say they are learning from a specific person actually share knowledge with that person.

## Discussion

To our knowledge, this is the first application of ERGM to medicinal plant knowledge networks and first test of how isolated nodes, directed triangles, intermediate connectivity, or reciprocity affect ecological knowledge networks. We analyzed medicinal knowledge sharing to test whether prestige and homophily affect network structure and learning patterns. We find multiple pieces of evidence for all four hypotheses, and these patterns are consistent across all five types of medicinal knowledge (Table 2, M3-7). These data have limitations, but we propose the following generalizations for medicinal plant knowledge social learning networks:

- Prestige reduces the number of edges, intermediate connectivity, and reciprocity, but increases inequality of node popularity and node isolation

- Homophily increases the amount of ISP triangles (Fig 1 H2c)

- In the context of prestige, learning patterns are simultaneously affected by knowledge, gender, and age

- In the context of homophily, learning patterns simultaneously affected by close kinship (direct ancestors), general kinship (nearby houses), marriage, co-residence, and gender

- Cultural marriage practices affect the hypothesized effects of kinship homophily and age-based prestige

- The strongest influences on network formation are the prestige of having unusual medicinal knowledge and the restriction of sharing knowledge to maintain prestige. However, homophily of close kin, general kin, spouses, and village residents also have large effects on knowledge sharing relationships

Together, these results suggest that prestige and homophily both affect social learning but have distinct functions.

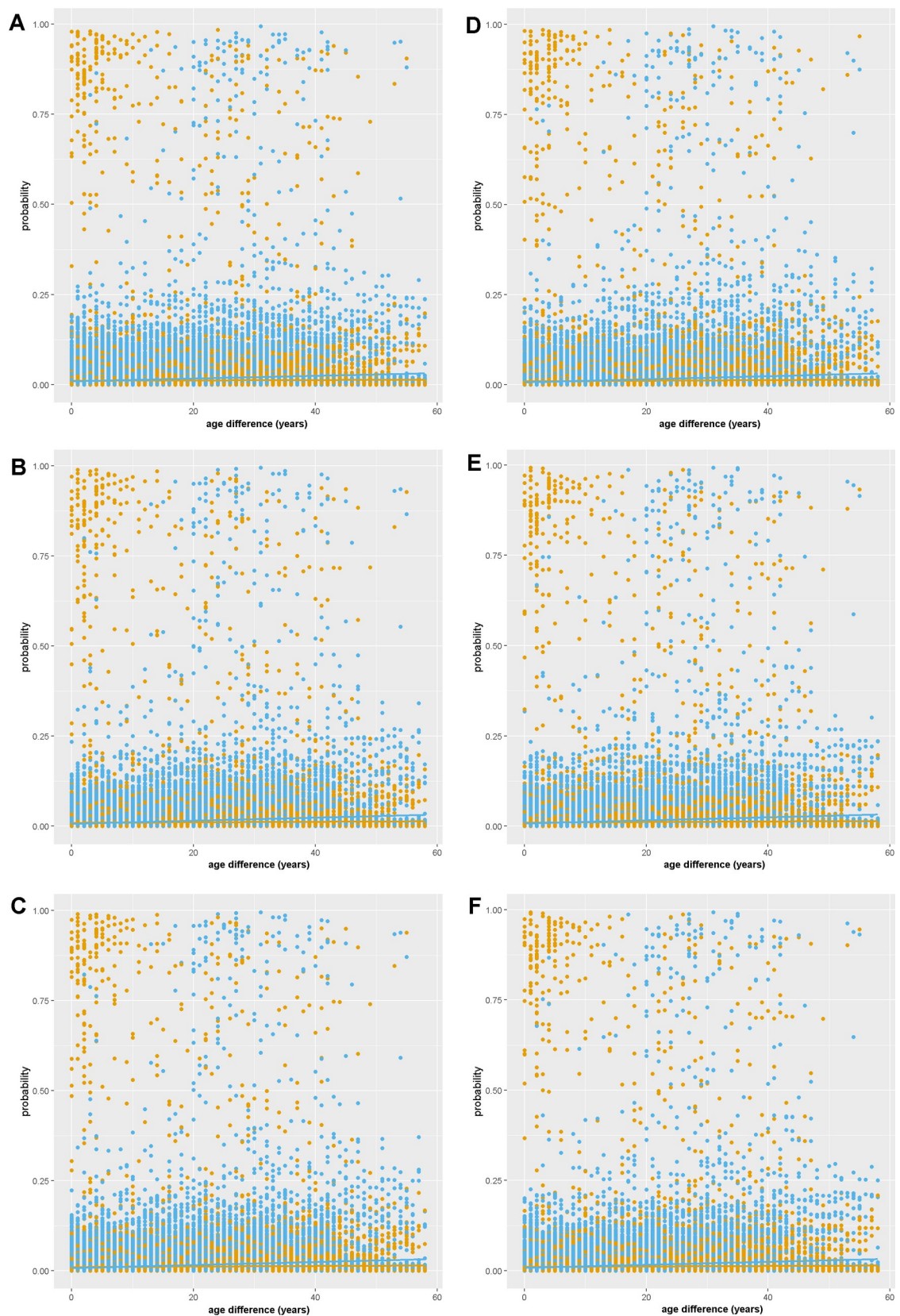

**Fig 5. Probability of an edge forming between each of the 303 people in the network, based on the age difference of each pair.** Dots representing the probability of each possible edge calculated from exponential random graph models are orange for man-woman pairs and blue for same-gender pairs. Key: A = M2, B = M3, C = M4, D = M5, E = M6, F = M7.

To our knowledge, this is the first ERGM of social learning to confirm that knowledge is shared between a learner and the person they learned. This confirms that self-reports are not overestimating the role of parents or other relationships in social learning due to social expectations [83], which is similar to previous findings that self-reports accurately represent knowledge sharing between parents and offspring [43]. A second reason to believe that social norms did not affect who learners cited is the fact that many participants did not cite their parents, and several explicitly stated that their parents did not know about medicine or chose not to share medicinal knowledge with them. A different potential source of bias is the ability of participants to remember who taught them about plants at any point in their life. However, this study was designed so that participants were asked to name who they learned from immediately after looking at photographs and hearing the names of each plant, which can help trigger memories related to the plants [65]. Additionally, without prompting, many participants specifically listed which plants each person taught them about. These factors both suggest that memory bias had only minimal effect in this study.

Although we show that prestige increases popularity of men in this network, we are reluctant to generalize this result for two reasons. First, there are no significant gendered differences in medicinal knowledge a global scale, only continental and national scales [84]. Therefore, observed differences in medicinal knowledge and its accompanying prestige are limited by cultural patterns of gender roles. Our results are consistent with patterns of power and knowledge in Melanesia [41, 42], and are likely generalizable to other patrilineal cultures that associate medicine with prestige. Second, our findings may have been influenced by the fact that all data were collected by a male [85]. In a region where contact between men and women is very regulated, it may have been easier for men to share more openly and completely [46]. Other possible biases in data collection include hesitancy to share cultural knowledge with an outsider, or fatigue based on interview length. In addition, our data do quantify the number of plants or remedies learned from each person; additional research is needed to understand which relationships in a network tend to be the source of the largest quantity of information or certain types of information (i.e. remedies for specific illnesses such as infections) [43].

The cultural roles of men and women in the study region also explain our two non-significant results- kinship homophily and age prestige. The cluster of high probability man-woman pairs that are not both from the main kinship group (Fig 4) suggests that marriage between kinship groups may mask a weak positive effect of kinship homophily. The cluster of high probability man-woman pairs with <10 years difference in age (Fig 5) suggests that spousal homophily may mask a weak positive effect of age prestige. Future research is needed to establish whether the stronger effect of kinship homophily and age prestige in previous studies [19, 40] is because we account for additional factors such as network configuration and house proximity, or whether Melanesian cultures share medicinal knowledge less freely with relatives or peers [15].

One of the major challenges in network analysis is defining network boundaries [86]. In our network, a large proportion of learning took place with people who were not interviewed: 35% of the edges in the global open network are between informants (Fig 3), and 65% of them are between informants and people who were not interviewed (S2 Fig), such as relatives in other villages, colleagues from school or temporary work in urban areas, and deceased persons. Because we could not collect personal information from these people, their effect on prestige

and homophily is uncertain. Therefore, our results may not reflect the exchanges with individuals outside the four study villages. However, most people reported as sources of knowledge were older relatives that they had lived with (e.g., parents, uncles, aunts, grandparents), so including these persons in the model would likely confirm our conclusions about the importance of age, kinship, and proximity.

We are confident that our results are valid within the study area because our sampling effort was high (>90%; S1 Table) and rarefaction curves show >99% coverage of medicinal plant species cited by study participants (S1 Fig). We are also confident that study participants learned most of their medicinal plant knowledge via social learning because 97% of participants reported learning about medicinal plants exclusively other people. Study participants never mentioned learning from mass media influences (e.g., books, TV, internet). In all villages, cell phone service is intermittent and phones owned by residents are unable to use the internet. Video players are common, but only play downloaded content (i.e., movies). Radios and books are uncommon and do not include information about local plants. Participants only mentioned two methods of learning about plants through direct experimentation- sensory information and dreams. In this region, it is not uncommon for people to dream about specific plants, then experiment with what illness to treat with the plant [56]. However, only 5 participants reported learning through dreams, and only 2 participants reported learning via sensory characteristics (e.g., smell, taste).

## Social learning implications

Our results are an important contribution to the limited number of studies that combine demographic data and social network metrics to test hypotheses of social learning in small-scale societies, especially for ecological knowledge [23, 87]. Previous research has established that knowledge can be shared in three directions: vertically (from parents or grandparents), obliquely (from other elders), or horizontally (from peers) [87]. Previous research has also established that the choice of medicinal plant teacher can be associated with certain demographic variables [15, 19, 39, 40, 43]. Our results expand on this foundation by specifying which types of people are most likely to share knowledge in a particular direction. For example, we show that for vertical and oblique teachers tend to be men who are knowledgeable about medicinal plants, which expands on previous findings that people prefer to learn about medicinal plants from knowledgeable men [40] or women [15], depending on the cultural system. Vertical and oblique teacher-learner pairs also tend to be same-gender relatives within the same village, a combination which is confirmed by previous research [19]. We also show that there is no general pattern of horizontal learning; however, spouses are more likely to learn from each other, especially from the husband or the spouse that has more or unusual knowledge, and peers who are same-gender relatives in the same village are also more likely to learn from each other. Previous research confirms this combination entirely [19] and for knowledgeable same-gender co-residents [15].

For more insight on the mechanisms of social learning, future research should test whether certain directions of learning are associated with different learning processes. Potential situations for social learning suggested by the first author's participant observation include vertical learning via intentional teaching, horizontal learning via observation and imitation, and oblique learning via seeking specific knowledge. Another area for future study is to test how social learning networks change over time and whether certain kinds of knowledge are more likely to be learned or forgotten at certain ages. These and other temporal network questions will require longitudinal approaches, which have rarely been applied to medicinal knowledge (but see [43]).

## Biocultural conservation implications

Our results highlight several mechanisms of social learning with implications for conserving medicinal plant knowledge and other types of ecological knowledge. Importantly, we provide the first evidence of which conservation strategies related to prestige and homophily are most likely to have the greatest impact on ecological knowledge networks. However, any adaptation of ecological knowledge and practices to continuing social and ecological change must defer to Indigenous governance rights and practices [32]. Collaborating with ecological knowledge practitioners is essential for implementing conservation policy but also improves conservation outcomes [27, 88, 89].

Although prestige predicts several aspects of knowledge sharing network configuration and node similarity, prestige is based on the value of ecological knowledge, which can change based on the attitudes and perceptions of a society [13]. Our results suggest that decreasing prestige of medicinal knowledge could cause knowledge to be shared more freely, which would enhance resiliency of the knowledge network by increasing the number of edges and equality of edge distribution [53, 54]. However, loss of prestige could also decrease interest in learning, remembering, or sharing ecological knowledge, which would lead to the breakdown of personal knowledge sharing [90, 91] or rise of learning from non-human sources (e.g., books, internet) [92].

These scenarios suggest several avenues to conserve transmission of prestige-associated knowledge, such as medicinal plant knowledge. One method is to increase documentation of ecological knowledge so that knowledge can be learned from media sources, such as books or recordings [93]. However, the efficacy of this approach depends on empowering Indigenous and local people groups to choose which research questions are asked and how data are shared [32, 94, 95]. A second method is to increase social and institutional support for the value of ecological knowledge. Examples include incorporating ecological knowledge or local languages in formal education curricula, adopting school schedules that accommodate significant social and ecological events, and supporting complimentary use of herbal treatments at hospitals [96–98].

Interestingly, differences in the four research villages suggest that current educational, health, and economic systems in urban Solomon Islands are not supporting local knowledge. Kolofi is the only village adjacent to a gravel road which improves access to larger towns, markets, schools, and hospitals. Kolofi is also the least dense, least centralized, and most disconnected village; it also has the lowest average medicinal knowledge per resident (S1 Table). These results confirm previous research, which links community market integration and migration to lower network density and less ecological knowledge, including medicinal plant knowledge [42, 99, 100].

Our results also demonstrate how homophily predicts similarities between people who share knowledge. The strongest homophily variable in our analysis is spouses, which has interesting implications in the context of rising intermarriage both in Solomon Islands [101] and globally [102]. Intermarriage of kinship groups has always existed in North Malaita [46], but inter-island marriages are increasingly common. Spouses who enter the village may therefore serve as brokers of knowledge between physically and culturally distant groups [103], which may increase knowledge diversity and innovation [53]. However, because ecological knowledge also depends on resource availability, spouses from regions that are ecologically distinct will not have access to the plants that they know and cannot introduce their knowledge to other people [25]. Although spouses may transport and cultivate plants from their birth home, the limited availability of these plants is unlikely to affect other families in the village [104]. Thus, the global rise in intermarriage may increase ecological knowledge if the social groups

are in regions with similar plants but have no effect on ecological knowledge diversity if the regions are ecologically distinct.

Another strong homophily variable in our results, co-residence, suggests another way to increase the resilience of medicinal plant knowledge. Globally, urban populations are increasing, including populations of Indigenous peoples [105]. As urbanization increases population density, there may be more opportunities for sharing ecological knowledge. However, one of the main challenges for sharing ecological knowledge in urban areas is the lack of natural resources [25]. Thus, government policies that increase urban plant quantity, diversity, non-toxic management, legality of sustainable harvest, and accessibility, such as landscaping, urban forests, school gardens, and community gardens, will support continued transmission of ecological knowledge in a future of increasing urbanization [106, 107].

## Conclusions

Our study is the first test of whether hypotheses based on prestige and homophily predict both network structure and social learning patterns. By using ERGMs, we control for potentially confounding factors such as geographic proximity, network configuration, and stated versus actual knowledge sharing. Our work reveals that prestige and homophily have distinct effects on specific social learning strategies and knowledge types. By testing mechanisms of ecological knowledge sharing and the flow of medicinal plant information through social networks, this research suggests how ecological knowledge systems can be amended or supported to improve biocultural conservation. It also paves the way for quantifying the flow of medicinal knowledge based on cultural context and social networks. In an era of unprecedented global change, understanding social learning and ecological knowledge transmission is essential to support biocultural sustainability and enhance pharmaceutical development.

## Supporting information

**S1 Fig. Medicinal plant knowledge sampling adequacy rarefaction curves demonstrate >99% coverage of medicinal plant species cited by study participants.** Curves show coverage-based rarefaction (solid lines) and extrapolation (dashed lines). 95% confidence intervals (the shaded area associated with each curve) were estimated using 500 bootstrap replicates. Shannon and Simpson diversity account for both richness and evenness of the species cited. Species richness, Shannon diversity, and Gini-Simpson diversity ranged from 161-218, 92.2–139.9, and 71.6–108.0, respectively. Key: Shannon diversity index = exponential of Shannon entropy, Gini-Simpson diversity index = inverse Simpson concentration, Key: B = Binaoli village, I = Irobulu village, K = Kolofi village, L = Lagoe village.
(TIF)

**S2 Fig. Fruchterman-Reingold representation of the open medicinal plant knowledge network among the 303 participants and 729 total nodes.** Dots representing participants are colored according to village (Binaoli = orange, Kolofi = green, Irobulu = blue, Lagoe = yellow). Dots representing people outside of the network (not interviewed because deceased or in other villages) are white. Arrows connecting dots represent flow of knowledge shared from one person to another.
(TIF)

**S3 Fig. Correlations of variables used in mixed-effect models of network centrality, demographic, and medicinal plant knowledge.** N = 303 people. Panels on the lower left display Pearson correlation coefficients. Panels on the upper right show circles whose size represents Pearson correlation coefficients. Weaker correlations are displayed in more transparent font

and with smaller circles. Key: $^*p<0.1$; $^{**}p<0.05$; $^{***}p<0.01$.
(TIF)

**S4 Fig. Goodness-of-fit plots for the ERGM of M1 (configuration variables).** Solid lines represent the distributions of the observed statistics, and box plots summarize their distributions based on 100 simulated networks.
(TIF)

**S5 Fig. Goodness-of-fit plots for the ERGM of M2 (M1 + edge & node variables).** Solid lines represent the distributions of the observed statistics, and box plots summarize their distributions based on 100 simulated networks.
(TIF)

**S6 Fig. Goodness-of-fit plots for the ERGM of M3: (M2 + # uses known).** Solid lines represent the distributions of the observed statistics, and box plots summarize their distributions based on 100 simulated networks.
(TIF)

**S7 Fig. Goodness-of-fit plots for the ERGM of M4 (M2 + # species known).** Solid lines represent the distributions of the observed statistics, and box plots summarize their distributions based on 100 simulated networks.
(TIF)

**S8 Fig. Goodness-of-fit plots for the ERGM of M5 (M2 + # illnesses known).** Solid lines represent the distributions of the observed statistics, and box plots summarize their distributions based on 100 simulated networks.
(TIF)

**S9 Fig. Goodness-of-fit plots for the ERGM of M6 (M2 + use uniqueness).** Solid lines represent the distributions of the observed statistics, and box plots summarize their distributions based on 100 simulated networks.
(TIF)

**S10 Fig. Goodness-of-fit plots for the ERGM of M7 (M2 + species uniqueness).** Solid lines represent the distributions of the observed statistics, and box plots summarize their distributions based on 100 simulated networks.
(TIF)

**S11 Fig. ERGM MCMC diagnostics for M1: Configuration variables.**
(TIF)

**S12 Fig. ERGM MCMC diagnostics for M2: M1 + edge & node variables.**
(TIF)

**S13 Fig. ERGM MCMC diagnostics for M3: M2 + # uses known.**
(TIF)

**S14 Fig. ERGM MCMC diagnostics for M4: M2 + # species known.**
(TIF)

**S15 Fig. ERGM MCMC diagnostics for M5: M2 + # illnesses known.**
(TIF)

**S16 Fig. ERGM MCMC diagnostics for M6: M2 + use uniqueness.**
(TIF)

**S17 Fig. ERGM MCMC diagnostics for M7: M2 + species uniqueness.**
(TIF)

**S1 Table. Summary information for closed network of research villages in North Malaita, Solomon Islands.**
(DOCX)

**S1 File. Interview guides.**
(DOC)

**S1 Appendix. Medicinal plant list–alphabetical by Latin name.**
(DOCX)

**S2 Appendix. List of illnesses and illness categories (based on international classification of primary care).**
(DOCX)

## Acknowledgments

We are thankful for the assistance of the communities where field research was conducted: Kolofi, Binaoli, Irobulu, and Lagoe. We thank Kasey E. Barton, Tamara Ticktin, Mark Merlin, Alex Golub, and four anonymous reviewers for their comments that greatly improved the manuscript.

## Author Contributions

**Conceptualization:** Matthew O. Bond, Orou G. Gaoue.

**Data curation:** Matthew O. Bond.

**Formal analysis:** Matthew O. Bond.

**Funding acquisition:** Matthew O. Bond.

**Investigation:** Matthew O. Bond.

**Methodology:** Matthew O. Bond, Orou G. Gaoue.

**Project administration:** Matthew O. Bond.

**Resources:** Matthew O. Bond.

**Software:** Matthew O. Bond.

**Supervision:** Matthew O. Bond, Orou G. Gaoue.

**Validation:** Matthew O. Bond.

**Visualization:** Matthew O. Bond.

**Writing – original draft:** Matthew O. Bond.

**Writing – review & editing:** Matthew O. Bond, Orou G. Gaoue.

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
