## [Decision Letter · Decision Letter 0]

30 Jun 2020

PONE-D-20-15541

Prestige and Homophily Predict Network Structure for Social Learning of Medicinal Plant Knowledge

PLOS ONE

Dear Dr. Bond,

Thank you for submitting your manuscript to PLOS ONE. After careful consideration, we feel that it has merit but does not fully meet PLOS ONE’s publication criteria as it currently stands. Therefore, we invite you to submit a revised version of the manuscript that addresses the points raised during the review process.

Thank you very much for the opportunity to edit your manuscript. The reviewers did a good job of analyzing. I ask you to pay special attention to reviewers 1 and 4.

We look forward to receiving your revised manuscript.

Kind regards,

Prof. Dr. Ulysses Paulino Albuquerque

Academic Editor

PLOS ONE

Journal Requirements:

2. Please include additional information regarding the interview guide or script used in the study and ensure that you have provided sufficient details that others could replicate the analyses. For instance, if you developed a guide as part of this study and it is not under a copyright more restrictive than CC-BY, please include a copy, in both the original language and English, as Supporting Information.

4.We note that [Figure(s) 1] in your submission contain [map/satellite] images which may be copyrighted. All PLOS content is published under the Creative Commons Attribution License (CC BY 4.0), which means that the manuscript, images, and Supporting Information files will be freely available online, and any third party is permitted to access, download, copy, distribute, and use these materials in any way, even commercially, with proper attribution. For these reasons, we cannot publish previously copyrighted maps or satellite images created using proprietary data, such as Google software (Google Maps, Street View, and Earth). For more information, see our copyright guidelines: http://journals.plos.org/plosone/s/licenses-and-copyright.

You may seek permission from the original copyright holder of Figure(s) [1] to publish the content specifically under the CC BY 4.0 license. 

If you are unable to obtain permission from the original copyright holder to publish these figures under the CC BY 4.0 license or if the copyright holder’s requirements are incompatible with the CC BY 4.0 license, please either i) remove the figure or ii) supply a replacement figure that complies with the CC BY 4.0 license. Please check copyright information on all replacement figures and update the figure caption with source information. If applicable, please specify in the figure caption text when a figure is similar but not identical to the original image and is therefore for illustrative purposes only.

Reviewers' comments:

Reviewer's Responses to Questions

**Comments to the Author**

1. Is the manuscript technically sound, and do the data support the conclusions?

Reviewer #1: Yes

Reviewer #2: Yes

Reviewer #3: Yes

Reviewer #4: Yes

2. Has the statistical analysis been performed appropriately and rigorously? 

Reviewer #1: Yes

Reviewer #2: Yes

Reviewer #3: Yes

Reviewer #4: Yes

3. Have the authors made all data underlying the findings in their manuscript fully available?

Reviewer #1: Yes

Reviewer #2: Yes

Reviewer #3: Yes

Reviewer #4: Yes

4. Is the manuscript presented in an intelligible fashion and written in standard English?

Reviewer #1: No

Reviewer #2: Yes

Reviewer #3: Yes

Reviewer #4: Yes

5. Review Comments to the Author

Reviewer #1: The study intitled “Prestige and homophily predict network structure for social learning” is very interesting since it contributes to the understanding of the factors that can influence the structure of the knowledge sharing network about medicinal plants in human groups. However, different aspects throughout the text need to be made clearer. Below, I highlight these points.

In lines 65-66, the authors define social learning as “…the ability to transmit information and ideas between generations". However, for the works cited by the authors in the introduction, social learning would involve processes linked to learning information through observation, interaction with other members of the social group or contact with products built by other individuals, not necessarily between generations, but it can occur between individuals belonging to the same generation in a group. In this sense, I believe that this passage can be modified. In addition, social learning strategies do not only involve imitation, as suggested by the text in line 76, which would also need to be relativized.

In lines 68-69, the following excerpt: “Because humans are a single species, the key to this global domination is not biological adaptation, but the related concept of cultural adaptation..." can be rethought. Several studies suggest that cultural practices have favored the recent positive selection of certain genes in humans as well as biological factors may also influence cultural evolution (O'Brien & Laland 2012; Mesoudi 2015; Altman & Mesoudi 2019). Perhaps it is the case to adjust this stretch, incorporating that both biological and cultural adaptations and the interactions between these (biocultural) factors are relevant to our understanding of human success in different environments.

*Altman, A.; Mesoudi, A. 2019. Understanding agriculture within the frameworks of cumulative cultural evolution, gene-culture co-evolution, and cultural niche construction. Human Ecology 47: 483-497.

*Mesoudi, A. 2015. Cultural evolution: a review of theory, findings and controversies. Evolutionary Biology 43: 481-497.

*O'Brien, M.J.; Laland, K.N. 2012. Genes, culture, and agriculture: na example of human niche construction. Current Anthropology 53: 434-470.

In the paragraph on lines 76-82, the authors briefly present the possible role of prestige and homophily in explaining social learning patterns. However, the authors indicate different theories for each factor that are not clearly presented, which can confuse the reader about the different theoretical scenarios involved. In this case, it is possible to place prestige and homophily as distinct types of model-based social learning biases, based on the theoretical framework of cultural evolution (see Mesoudi 2015). The indication and brief presentation of the theory of cultural evolution can help in the theoretical understanding to which the factors presented are inserted.

*Mesoudi, A. 2015. Cultural evolution: a review of theory, findings and controversies. Evolutionary Biology 43: 481-497.

In line 83, the authors highlight the following: "One of the most fundamental types of social learning for human survival is ecological knowledge..." This passage may seem confusing to the reader, as the types of social learning can involve the different mechanisms and social learning strategies, but not necessarily the knowledge domains that are involved in these strategies. So, it may be interesting to substitute this excerpt for “One of the most fundamental knowledge domains for…”.

In line 133, the authors indicate “social learning theories”. This passage can confuse the reader. In this case, they are not necessarily theories, but different processes or biases of social learning.

In the topic “Theoretical framework”, during the presentation of the hypotheses, it might be interesting to clearly indicate why a hypothesis about the influence of homophily on the knowledge sharing network configuration was not addressed in the text. At the beginning of this topic, the authors point out that the work investigate "how homophily and prestige...affect....specific patterns of network connections", but does not address a hypothesis that relates homophily to the network configuration.

In the topic “Fieldwork site”, I missed more explanations about the socioenvironmental characteristics of the groups studied, such as the ecosystems to which they belong, proximity to urban centers, beliefs, the social structure (and its variation between each group), the organization of people in different social roles, geographical distance between these groups. This can even help the reader to understand why certain sociocultural variables were analyzed in this study.

In the topic “ERGM analysis”, there are two repeated parts of the text (lines 297-299 and lines 300-301). I suggest deleting one of the parts.

In the results, in the topic “Hypothesis 1” it is still not clear to me why only the prestige is being considered in the hypothesis of the network configuration, considering that there are results on the effect of homophily. Perhaps the topic title needs to consider the effect of homophily.

In lines 421-423, the authors write the following excerpt “...the probability of an edge forming between two people whose age difference is <10 years is higher for man-woman pairs than for same-gender pairs, indicating that nonsignificance of age prestige may be related to cultural gender roles." I was confused in this part of the text because what seems to be being discussed here is a factor linked to homophily, that is, people with close ages. If it is the case to change this part of the text, some points of the discussion about this finding may also be modified (lines 448 and 480).

In the topic “Social learning implications”, the authors point out that "Our results address a lack of prior research on the process and patterns of social learning in small-scale societies, especially in learning ecological knowledge [21,76]. Previous research has established that knowledge can be shared in three directions: vertically (from parents or grandparents), obliquely (from other elders), or horizontally (from peers) [76]." However, there are studies that have investigated and discussed the sharing of local ecological knowledge (such as knowledge of medicinal plants) between people and the biases linked to social learning in small-scale societies (see, for example, Henrich & Broesch 2011; Salpeteur et al. 2015; Brito et al. 2019; Santoro et al. 2020). In particular, the work of Henrich & Broesch (2011), also cited by the authors of the manuscript, investigates learning biases (such as prestige, sex and age) involving local ecological knowledge through network analysis. The mentioned works indicate the possible biases that may be behind the processes of transmission of local ecological knowledge. It would be interesting, then, to discuss the findings of the present manuscript on this topic with previous studies that have investigated biases related to knowledge sharing about medicinal plants. If this adjustment is made on this topic, it will also be necessary to adjust the conclusion section on lines 565-566, in which the authors state "Our study is the first test of how prestige and homophily relate to network structure and social learning patterns" and an excerpt from the introduction of the manuscript, on lines 95-97. In this last point about the text of the introduction, the originality and relevance of the research can be rethought.

*Brito, C.C.; Ferreira Júnior, W.S.; Albuquerque, U.P.; Ramos, M.A.; Silva, T.C.; Costa-Neto, E.M.; Medeiros, P.M. 2019. The role of kinship in knowledge about medicinal plants: evidence for context-dependent model-biases in cultural transmission? Acta Botanica Brasilica 33:370-375.

*Henrich, J.; Broesch, J. 2011. On the nature of cultural transmission networks: evidence from Fijian villages for adaptive learning biases. Philosophical Transactions of the Royal Society B 366:1139-1148.

*Salpeteur, M.; Patel, H.; Balbo, A.L.; Rubio-Campillo, X.; Madella, M.; Ajithprasad, P.; Reyes-García, V. 2015. When knowledge follows blood. Kin groups and the distribution of traditional ecological knowledge in a community of seminomadic pastoralists, Gujarat (India). Current Anthropology 56:471-483.

*Santoro, F.R.; Chaves, L.S.; Albuquerque, U.P. 2020. Evolutionary aspects that guide the cultural transmission pathways in a local medical system in Northeast Brazil. Heliyon 6:e04109.

Reviewer #2: The manuscript is clear, objective and presents important theoretical and methodological contributions. Some considerations:

Line 46 - I do not agree that there is a limited understanding of the processes that define the transmission of local ecological knowledge. There are several studies that analyze different variables, such as prestige, age and environmental condition.

Line 76 - according to Hoppitt and Laland (2008) and Rendell et al. (2010) there are many social learning processes, in addition to “imitation”, such as “pavovlian conditioning”, “contextual imitation” and “teaching”. So, I suggest that the authors modify this sentence.

Hoppitt, W. & Laland, K. N. 2008 Social Processes Influencing Learning in Animals: A Review of the Evidence. Advances in the Study of Behavior 38: 105-165; Rendell, L .; Forgaty, L. & Laland, K. N. 2010. Rogers ’paradox recast and resolved: population structure and the evolution of social learning strategies. Evolution 64: 534–548.

Line 95-97: I suggest the authors to read “HEINEBERG, Marian Ruth and HANAZAKI, Natalia. Dynamics of the botanical knowledge of the Laklãnõ-Xokleng indigenous people in Southern Brazil. Acta Bot. Bras. [online]. 2019, vol.33, n.2, pp.254-268. Epub June 19, 2019. ISSN 1677-941X. https://doi.org/10.1590/0102-33062018abb0307. ” In this paper, the authors conclude, for example, that the elders, highly prestigious people in indigenous societies, are central to the structuring of transmission networks.

Line 468-478: The social network was built considering the models mentioned by each informant at the end of interview 2. However, it is possible that the same person may have been a model for the knowledge of more than one medicinal plant. In this case, the number of events established between a model and an leaner is also a fundamental variable to understand the structure of the transmission of knowledge in a social network. I believe that the authors need to briefly discuss how the absence of this information needs to be considered when analyzing the data.

Line 565-566: As I stated earlier, I believe the paper HEINEBERG, Marian Ruth and HANAZAKI, Natalia. Dynamics of the botanical knowledge of the Laklãnõ-Xokleng indigenous people in Southern Brazil. Acta Bot. Bras. [online]. 2019, vol.33, n.2, pp.254-268. Epub June 19, 2019. ISSN 1677-941X, is original in presenting some data about the prestige and structure of the social network.

Reviewer #3: The authors show how prestige and homophilia are strong structurers of social learning patterns, for this they use networking tools to visualize the complexity of ethnobotanical systems using abstract models. In this way, the approach allows detecting interactions and components not observable with the naked eye, revealing highly relevant aspects in ethnobotanical theory. The work stands out for its logical coherence and supports each of its hypotheses and reflections with a theoretical framework of evolutionary ethnobiology and network theory, clearly reflected in the reference bibliography. The data is surveyed using current ethnobotanical methodological techniques, complies with the guidelines of the code of ethics and the analyzes have adequate statistical support.

The authors could discuss the vulnerability of a knowledge system with a distribution focused on few agents, Castiñeira Latorre et al. 2020 discusses hipotetical scenarios in botanical medical systems, it removes nodes for experimentally with varying degrees of connection and evaluates the robustness of the system. Finally, I highlight the global nature of the work and I thank the authors for their great contribution to the development of the knowledge area.

Reviewer #4: This is a very interesting article about social learning models. Combining semi-structured interviews, participant observation, checklist interviews, and a detailed social network analysis, authors assess different hypotheses about the social learning processes that shape medicinal plant knowledge across four villages in Solomon Islands. Hypothesis are well defined and presented, the text is clear and easy to follow. Figures and Tables are very informative. While I support the publication of this paper, there are few issues needed to be addressed beforehand. In general, the manuscript needs a larger cultural context, and a discussion that must go beyond of the exponential random graph models (ERGMs) findings:

Introduction

In my opinion, introduction section could be more focused in social learning models, containing other factors and experiences discussed in literature, including the importance of individual learning, mass media influence, gender, etc.

Fieldwork site

This section need to be enriched by socio cultural information about the people of the four communities of Salomon Islands. Are they indigenous communities? Is the same the importance of the medicinal plant use in the four communities? Which is the medical system in each community? What plants are mainly used? In what extend people depend on medicinal plants? Could also be of interest to know about the presence of biomedicine in local systems. I know that this is not the purpose of the manuscript, but a brief description could be of essential to understand the sociocultural context.

Methodology

Self-reports about people from who they learned about medicinal plants, has been object of criticism. It is argued that people tend to overvalue the role of parents in the learning process when asked from whom they learned any knowledge, overestimating vertical routes. Please discuss this limitation.

Discussion

Hypothesized patterns of knowledge sharing based on prestige and homophily are more common in the observed network than in randomly simulated networks. Authors should discuss other influencing factors (not included in these framework) that could affect the network structure and learning processes of medicinal plants in these communities: such as individual learning, age, mass media influences, the effect of people that are not included in the studied networks. Because the results are not reflecting the exchanges with individuals outside the study area who were not interviewed. In addition, in my opinion could be of interest to discuss more the differences among Kolofi, Binaoli, Irobulu, and Lagoe. It is possible to read some local voices about their medicinal plant learning process?

As the authors say, the fieldwork was very limited (and unique!, in my opinion!), that they cannot used it as a general model to other cultural systems, could you please discuss more about of this limitation?

Authors say that their results are not reflecting the exchanges with individuals outside the study area who were not interviewed. Please could you please discuss more about this limitation of the model?

What about the spread of innovations in these communities with the models found?

The idea about intermarriage can “obscure” certain aspects of prestige and homophily, in my opinion, must be explained in a different way, avoiding gender bias in the interpretation.

Biocultural implications

Authors consider that the flow of medicinal plant information through social networks in these communities have the same behavior that the flow of ecological knowledge information. I suspect that they are very different to make inferences about to improve biocultural conservation. Significant differences in governance processes of natural resources can be expected among networks experiencing structural differences in terms of density of relations, degree of cohesiveness, subgroup interconnectivity, and degree of network centralization, but there are different rules in indigenous communities that shapes communal decisions, that are very different to individual decisions. Please, clarify this topic.

6. PLOS authors have the option to publish the peer review history of their article (what does this mean?). If published, this will include your full peer review and any attached files.

Reviewer #2: No

Reviewer #4: No

---

## [Author Response · Author response to Decision Letter 0]

1 Sep 2020

To preserve formatting, my response to the editor's comments and the reviewer comments is in the "response to reviewers" document

---

## [Editor Report · Decision Letter 1]

4 Sep 2020

Prestige and Homophily Predict Network Structure for Social Learning of Medicinal Plant Knowledge

PONE-D-20-15541R1

Dear Dr. Bond,

We’re pleased to inform you that your manuscript has been judged scientifically suitable for publication and will be formally accepted for publication once it meets all outstanding technical requirements.

Kind regards,

Prof. Dr. Ulysses Paulino Albuquerque

Academic Editor

PLOS ONE
---

## [Editor Report · Acceptance letter]

14 Sep 2020

PONE-D-20-15541R1 

Prestige and Homophily Predict Network Structure for Social Learning of Medicinal Plant Knowledge 

Dear Dr. Bond:

I'm pleased to inform you that your manuscript has been deemed suitable for publication in PLOS ONE. Congratulations! Your manuscript is now with our production department. 

Kind regards, 

on behalf of

Dr. Ulysses Paulino Albuquerque 

Academic Editor

PLOS ONE